# Does Genetic Variation in Detoxification Capacity Influence Hepatic Biomarker Responses to a Liver Support Supplementation Regimen?

**DOI:** 10.3390/ijms262010209

**Published:** 2025-10-20

**Authors:** Markus Schauer, Susanne Mair, Michael Keiner, Christian Werner, Florian Kainz, Mohamad Motevalli

**Affiliations:** 1Department of Sport Science, German University of Health & Sport (DHGS), 85737 Ismaning, Germany; michael.keiner@dhgs-hochschule.de (M.K.); florian.kainz@dhgs-hochschule.de (F.K.); 2VerticalMed Tyrol, 6065 Thaur, Austria; susanne@verticalmedtyrol.com; 3University Institute of Schaffhausen, CH-8200 Schaffhausen, Switzerland; christian.werner@eoe.univeristy; 4Department of Sport Science, University of Innsbruck, 6020 Innsbruck, Austria

**Keywords:** genetic predisposition, oxidative stress, redox balance, hepatic metabolism, glucuronidation, α-lipoic acid, glutathione, N-acetylcysteine, nutrigenomics, precision nutrition

## Abstract

Genetic polymorphisms contribute to inter-individual variation in liver detoxification, influencing susceptibility to exposures and responses to interventions. While urinary biomarkers reflect Phase II activity, the impact of genotype on supplementation response remains unclear. In a pilot, prospective, open-label cohort study, 30 Austrian adults completed an 8-week multi-ingredient liver support supplementation regimen (e.g., glutathione, N-acetylcysteine, α-lipoic acid). First-morning urine was collected at baseline and follow-up for measurement of D-glucaric acid, mercapturic acids, and creatinine. Dried blood spots were genotyped for polymorphisms in detoxification genes (*CYP1A1*, *GSTM1*, *GSTP1*, *GSTT1*, and *NQO1*), and participants were stratified into normal (NDC) or limited (LDC) detoxification capacity groups. Adherence was monitored through logs, mid-study interviews, and product counts. The intervention led to modest reductions in body weight (−0.87 kg, *p* < 0.05) and BMI (−0.31 kg/m^2^, *p* < 0.05), and significant increases in urinary D-glucaric acid (*p* < 0.05) and mercapturic acids (*p* < 0.01), with consistent responses across detoxification genotype groups (*p* > 0.05). The pattern of biomarker responses based on clinically defined categories did not differ significantly between the study groups (adjusted odds ratio = 2.88, *p* > 0.05). The observed increases in urinary biomarkers and reductions in weight and BMI are consistent with potential modulation of detoxification pathways following liver support supplementation, independent of genetic polymorphisms influencing detoxification capacity.

## 1. Introduction

The liver is the principal organ for biotransformation and elimination of endogenous metabolites and xenobiotics [1]. Hepatic detoxification proceeds through coordinated multi-phase processes: Phase I enzymes (primarily cytochrome P450 monooxygenases) modify lipophilic substrates, Phase II enzymes (conjugating systems such as UDP-glucuronosyltransferases and glutathione S-transferases) increase water solubility, and Phase III transporters export modified metabolites for excretion [1,2,3,4]. Dysregulation at any of these steps can alter systemic exposure to electrophiles and reactive intermediates and thereby modulate disease risk and response to nutritional or pharmacologic interventions. Enhancing hepatic detoxification is therefore clinically relevant, as it can reduce the systemic burden of environmental toxins, support metabolic health, and help prevent chronic disease [5,6].

Inter-individual variation in detoxification is substantial and is driven by both environmental exposures and inherited genetic differences. Common polymorphisms and gene-deletions (notably *GSTM1* and *GSTT1* null genotypes) as well as functional single nucleotide variants in genes such as *GSTP1*, *NQO1*, and *CYP1A1* have been associated with altered enzyme activity or expression (potentially influenced by sex, hormonal status, and age), with implications for xenobiotic handling, oxidative stress, and disease susceptibility across populations [7,8,9,10]. A large-scale systematic review has documented the population frequency and potential phenotypic consequences of *GSTM1*/*GSTT1* variation and highlighted heterogeneity by ethnicity and exposure context [11]. These genetic differences provide a biologically plausible basis for inter-individual variability in response to detoxification-targeted interventions [12].

Since direct measurement of hepatic enzyme activity in humans is invasive and impractical in field studies, non-invasive urinary biomarkers are widely used as proxies of specific detoxification pathways [13]. Two complementary urinary readouts that are particularly informative for Phase II activity are D-glucaric acid, a metabolite associated with glucuronidation pathways and indicative of changes in glucuronide turnover and β-glucuronidase activity [14], and mercapturic acids, the end products of glutathione conjugation with electrophiles that serve as an integrated readout of GSH-dependent detoxification [15]. Both markers have well-established analytical methods (HPLC-UV, LC-MS/MS) and are increasingly used in exposure and intervention studies, where first-morning spot urine samples reliably capture intra-individual changes over short intervention windows and serve as suitable endpoints for quantifying shifts in biotransformation capacity [14,15,16,17].

In parallel with biomarker development, advances in nutrigenomics and precision nutrition suggest that targeted dietary or supplement interventions can modulate detoxification pathways, but responses are frequently genotype-dependent [6,18,19]. Trials and mechanistic studies show that compounds such as N-acetylcysteine (a precursor to glutathione), lipophilic or liposomal glutathione formulations, and α-lipoic acid (an antioxidant that can regenerate reduced glutathione and modulate redox signaling) can influence intracellular GSH pools and related detoxification outcomes [20,21,22,23]. Cruciferous-vegetable derivatives (e.g., indoles and isothiocyanates, with downstream metabolites such as diindolylmethane, DIM) are also known to induce Phase II enzyme expression, making them candidate agents for “liver support” interventions [24]. Despite the widespread availability of commercial liver support supplements, rigorous scientific validation of their efficacy remains limited. In particular, empirical data linking genetic detox profiles to measurable changes in hepatic biomarkers in response to such regimens is currently lacking.

Given this background, integrating genetic screening with targeted supplementation and sensitive urinary biomarkers offers a feasible precision-nutrition strategy to assess the modifiability of hepatic detoxification in real-world cohorts. This study aimed to investigate whether an 8-week liver support regimen can enhance urinary indicators of Phase II activity and whether inter-individual differences in genetic detoxification capacity shape the magnitude or pattern of these biomarker responses. This approach uniquely integrates genotyping, biomarker assessment, and adherence-controlled supplementation within a real-world cohort, enabling a focused evaluation of individualized nutritional strategies. It was therefore hypothesized that the intervention would significantly increase the urinary excretion of both mercapturic acids and D-glucaric acid, with a more pronounced response in these biomarkers demonstrated by individuals in the limited detoxification capacity group compared to those with normal capacity.

## 2. Results

The study included 30 participants (18 in LDC and 12 in NDC), with no significant differences observed between groups in terms of age, sex distribution, height, weight, or BMI (*p* > 0.05). All participants demonstrated full adherence to the supplementation protocol. Biomarkers of detoxification also showed comparable levels across groups at baseline: mean D-glucaric acid and mercapturic acid concentrations did not differ significantly (*p* = 0.9 and *p* = 0.4, respectively), nor did creatinine levels (*p* = 0.7). Categorical distributions of D-glucaric acid and mercapturic acid levels were similarly balanced between groups, with no statistically significant differences detected (*p* > 0.05). Baseline characteristics of the participants are summarized in Table 1. Figure 1 shows the distributions of measures in participants before and after the intervention stratified by detoxification capacity.

Overall, the intervention was associated with significant increases in the Phase II detoxification biomarkers D-glucaric acid and mercapturic acids, as well as significant reductions in body weight, BMI, and creatinine levels (*p* < 0.05), with no significant group differences for D-glucaric acid, mercapturic acids, body weight, or BMI between participants with limited and normal detoxification capacity (*p* > 0.05).

For body weight, the linear mixed-effects model showed a significant main effect of time, with participants showing a small but significant reduction from pre- to post-assessment (β = −0.87 kg, 95% CI [−1.60, −0.15], *p* = 0.020; F(1,30) = 6.65, *p* = 0.015). There was no significant main effect of group (β = 0.13, *p* = 0.974) and no time × group interaction (*p* = 0.600), indicating that the change in body weight did not differ between individuals with limited versus normal detoxification capacity. However, a significant effect of sex was observed, with males weighing more than females on average (β = 12.37 kg, 95% CI [3.92, 20.82], *p* = 0.006; F(1,30) = 8.94, *p* = 0.006). Model fit indices indicated that the fixed effects explained 24% of the variance (marginal R^2^ = 0.24), while the full model including random effects explained 99% of the variance (conditional R^2^ = 0.99). Residuals were normally distributed, supporting model assumptions. Table 2 summarizes the results of the linear mixed-effects model for body weight.

The model showed a significant main effect of time, with participants exhibiting a small but significant reduction in BMI from pre- to post-assessment (β = −0.31 kg/m^2^, 95% CI [−0.56, −0.06], *p* = 0.019; F(1,30) = 6.94, *p* = 0.013). No significant effects were observed for group (*p* = 0.572), sex (*p* = 0.528), or the time × group interaction (*p* = 0.610). Model fit indicated that fixed effects explained only 3% of the variance (marginal R^2^ = 0.03), while the full model explained nearly all variance due to strong random effects (conditional R^2^ = 0.99). Residuals were normally distributed, supporting model assumptions. Table 3 summarizes the results of the linear mixed-effects model for BMI.

For D-glucaric acid, the model indicated a significant main effect of time, with concentrations increasing significantly from pre- to post-assessment (β = 33.83, 95% CI [6.73, 60.93], *p* = 0.016). In addition, a significant effect of sex was observed, with males showing substantially lower levels compared to females (β = −43.19, 95% CI [−74.48, −11.89], *p* = 0.008). Neither the group effect (*p* = 0.563) nor the time × group interaction (*p* = 0.662) reached significance, indicating that detoxification capacity did not influence either baseline levels or changes over time. Model fit showed that fixed effects accounted for 22% of the variance (marginal R^2^ = 0.22), while the full model explained 47% of the variance (conditional R^2^ = 0.47). However, residuals deviated from normality (*p* = 0.01), suggesting that results should be interpreted with some caution. Table 4 summarizes the results of the linear mixed-effects model for D-glucaric Acid.

The model for mercapturic acids showed a clear effect of time, with levels significantly higher at post-assessment than at pre-assessment (β = 14.67, 95% CI [6.02, 23.32], *p* = 0.002). In contrast, no significant effects of group (*p* = 0.623), sex (*p* = 0.194), or the time × group interaction (*p* = 0.346) were observed, suggesting that increases in mercapturic acids over time were consistent across both detoxification capacity groups and sexes. The marginal R^2^ indicated that fixed effects explained 21% of the variance, while the full model explained 26% (conditional R^2^ = 0.26). Residuals deviated significantly from normality (*p* = 0.001), which again suggests caution in interpreting these results. Table 5 summarizes the results of the linear mixed-effects model for mercapturic acids.

For creatinine, the model showed a significant main effect of time, with levels decreasing significantly from pre- to post-assessment (β = −42.89, 95% CI [−81.67, −4.12], *p* = 0.031). Importantly, there was also a significant time × group interaction (β = 76.22, 95% CI [15.61, 136.83], *p* = 0.016), indicating that the magnitude of creatinine change differed between detoxification capacity groups. While the direction of this interaction suggests diverging trends between groups, group (*p* = 0.171) and sex (*p* = 0.092) main effects were not significant. Model fit indices indicated that fixed effects explained 15% of the variance (marginal R^2^ = 0.15), and the same amount was explained when including random effects (conditional R^2^ = 0.15). Residuals deviated from normality (*p* = 0.003), which limits the robustness of the conclusions. Table 6 summarizes the results of the linear mixed-effects model for Creatinine.

Analysis of changes in biomarker categories showed that improvements in D-glucaric acid levels occurred in 56% of the LDC group vs. 42% of the NDC group (*p* = 0.924). For mercapturic acid levels, improvements were observed in 13 of 18 LDC participants (72%) vs. 5 of 12 NDC participants (42%) (*p* = 0.165). Overall response to intervention showed dual improvement in 7 (39%) vs. 1 (8%) participants, single improvement in 7 (39%) vs. 8 (67%), stability in 2 (11%) vs. 2 (17%), and mixed response in 2 (11%) vs. 1 (8%) (*p* = 0.263; adjusted OR = 2.88, *p* = 0.150) for LDC and NDC groups, respectively. Table 7 summarizes the distribution of changes in biomarker categories and overall intervention responses across LDC and NDC groups.

Correlation analyses were conducted to explore the relationships among the changes (Δ, post–pre intervention) in five key variables: D-glucaric acid, mercapturic acids, body weight, BMI, and creatinine. The correlation matrix revealed several associations (Figure 2). D-glucaric acid exhibited moderate positive correlations with both body weight (*r* = 0.32) and BMI (*r* = 0.30), as well as with mercapturic acids (*r* = 0.30). Conversely, D-glucaric acid showed a moderate inverse correlation with creatinine (*r* = −0.48). Correlations between mercapturic acids and anthropometric parameters were weaker (*r* = 0.12–0.15), as was the relationship between mercapturic acids and creatinine (*r* = −0.15).

## 3. Discussion

The present study showed that following an 8-week liver support supplementation regimen, there were modest but statistically significant increases in urinary D-glucaric acid and mercapturic acid levels across all participants. Such increases may reflect improved hepatic clearance of potentially harmful compounds, which could hold clinical relevance for individuals with suboptimal detoxification efficiency. However, group differences (LDC vs. NDC) did not influence baseline or trajectory of change for D-glucaric acid or mercapturic acids, nor did the group × time interactions reach significance. These findings indicate an association between the supplementation period and increased urinary excretion of metabolites related to glucuronidation/deconjugation and glutathione-conjugation pathways, though causal attribution to the intervention cannot be confirmed and responses appeared independent of baseline detoxification capacity. In this context, however, it is important to note that increased urinary levels may reflect elevated excretion or turnover of conjugates rather than necessarily indicating enhanced detoxification capacity [25,26].

The significant post-intervention increase in D-glucaric acid (β ≈ +33.8) across participants suggests enhanced metabolic activity within the glucuronic acid and glucarate pathways, potentially reflecting increased release of glucuronide conjugates or reduced deconjugation stress. Given that D-glucaric acid inhibits β-glucuronidase and has been shown to mitigate liver toxicity by lowering reactive oxygen species, suppressing the accumulation of deconjugated toxins, and facilitating detoxification via glucuronide conjugates [14], the observed elevation may represent an adaptive upregulation of endogenous detoxification processes. Consistent with this pattern, mercapturic acid levels also increased significantly (β ≈ +14.7), indicating activation of the glutathione conjugation pathway or a related thiol-based mechanism in response to an increased burden of electrophilic and reactive xenobiotics. Since neither group, sex, nor group × time interaction was significant, the increase appears to have been uniform. This uniform response argues that environmental or exposure factors affected all participants similarly, or that the intervention is potent enough to override modest differences in detoxification capacity. This interpretation is consistent with evidence that mercapturic acid levels are sensitive indicators of electrophilic chemical exposure; however, because baseline concentrations vary substantially across individuals depending on diet, smoking status, environmental contact, and genetic or physiological variability [15], the observed uniformity may reflect common exposure patterns or the strong influence of the intervention.

The prespecified genotype stratification (LDC vs. NDC) did not predict baseline concentrations or trajectories of D-glucaric acid or mercapturic acids. Several non-exclusive explanations may account for this null effect. First, commonly studied polymorphisms such as *GSTM1* and *GSTT1* nulls can alter enzyme expression but often exert modest, context-dependent effects; occupational studies show genotype–phenotype associations for particular mercapturates that depend on the chemical, exposure magnitude, and compensatory activity of other conjugating enzymes [27,28,29]. Consistent with this, evidence shows that combining *CYP1A1*, *GSTM1*, and *GSTP1* variants into “high-risk” versus “low-risk” groups could reveal differences in benzo(a)pyrene metabolism and DNA adduct formation [30]. Second, the LDC/NDC binary classification may simply have failed to capture the most functionally relevant determinants of D-glucaric acid dynamics. For example, variation in UGT isoforms (*UGT1A* and *UGT2B* families), gene deletions such as *UGT2B17*, or interindividual differences in β-glucuronidase activity from host and microbial sources could be more determinative than GST variants [31,32,33,34]. Third, shared external exposures or a sufficiently potent inducing intervention may have stimulated Phase II responses across genotypes, masking genotype-specific differences; population biomonitoring studies frequently find that exposure intensity, diet, and other lifestyle factors explain more variance in mercapturate excretion than single detoxification polymorphisms [15,35]. The absence of a genotype effect on mercapturic acids responses therefore aligns with literature showing that GST genotype effects are inconsistent and often overshadowed by exposure intensity, co-exposures, and compensatory metabolic pathways [36,37]. Collectively, these considerations suggest that future precision-detoxification studies should combine functional phenotyping (enzyme activity assays), gut microbiome profiling to assess β-glucuronidase potential, repeated timed urine sampling with specific-gravity adjustment, and compound-specific mercapturate panels to resolve the mechanistic questions raised here. Importantly, despite the complexity of genotype–phenotype interactions, these findings highlight that adaptive Phase II detoxification responses operate at the system level, integrating genetic, microbial, and environmental influences to maintain metabolic homeostasis. Thus, while single-gene effects may appear modest, the cumulative impact of these pathways likely contributes meaningfully to broader physiological processes such as xenobiotic clearance, oxidative stress modulation, and overall metabolic resilience. To advance this field, future research should prioritize multi-gene panels, gene–environment interactions, and microbiome–genotype interplay to better characterize individual detoxification capacity and intervention responsiveness.

The observed sex difference in D-glucaric acid, with females exhibiting substantially higher urinary levels than males (β ≈ −43.2, *p* = 0.008), suggests that hormonal, metabolic, and enzyme activity differences contribute to its regulation. This pattern aligns with established sex differences in glucuronidation enzyme expression and activity; several human and mechanistic studies have documented sex-biased UGT expression (e.g., *UGT2B17*), and sex hormones such as estrogen and testosterone can modulate UGT regulation, producing clinically meaningful differences in glucuronidation rates between men and women [32,34]. Beyond enzyme expression, sex-dependent variations in microbiome composition and β-glucuronidase activity may further influence systemic glucuronide deconjugation, with hormone-driven shifts in microbiome communities potentially altering β-glucuronidase abundance and activity, thereby altering measured D-glucaric acid levels [31,38]. The gut microbial β-glucuronidase pool is highly variable across individuals and affects enterohepatic cycling of glucuronides and hormone metabolism [38,39], providing additional biological plausibility for the observed sex difference. Furthermore, sex hormone–dependent metabolism may interact with both hepatic UGT activity and microbial β-glucuronidase, creating a multifactorial regulatory network that enhances D-glucaric acid levels in females. In contrast, the absence of a sex effect in mercapturic acid levels suggests that its pathways are less influenced by sex-dependent factors and more directly tied to exposure to reactive species, with the rate-limiting steps showing minimal sex dependence in this cohort. Taken together, the data indicate that D-glucaric acid-associated pathways are particularly sensitive to sex-related biological factors, including UGT enzyme expression, hormone regulation, microbiome activity, and sex hormone–microbiome interactions.

Our data can be placed within a larger literature that links dietary constituents, supplements, and lifestyle to modulation of Phase II pathways. Preclinical studies and limited human data suggest that calcium-D-glucarate (and its metabolite 1,4-glucaro-lactone) inhibits β-glucuronidase and therefore can decrease enterohepatic reactivation of glucuronides, reducing tissue exposure to deconjugated toxins and influencing urinary D-glucaric acid levels [40]; animal work shows reduced tumorigenesis with D-glucarate supplementation [41]. Human data remain sparse and inconclusive, with few early Phase I studies but no definitive large RCTs [14,42]; our findings align with these preliminary human trials in demonstrating modulation of detoxification biomarkers, while extending them by capturing concurrent changes in both glucuronidation and glutathione-conjugation pathways. Similarly, NAC and other glutathione-enhancing strategies can augment GSH pools and plausibly alter mercapturic acid excretion and oxidative stress markers; clinical trials of NAC and glycine/NAC combinations have demonstrated modulation of glutathione homeostasis and downstream biochemical endpoints [43,44]. These mechanistic and interventional findings support the biological plausibility that an intervention or lifestyle shift during our study could simultaneously raise D-glucaric acid and mercapturic acids.

The modest but statistically significant reductions in body weight (≈−0.87 kg) and BMI (≈−0.31 kg/m^2^) suggest that participants experienced some degree of metabolic change during the study. Although these changes are small, they may have influenced detoxification biology independently. Weight loss has been associated with altered redox status, changes in glutathione and antioxidant enzyme levels, and shifts in hepatic metabolic enzyme expression (e.g., UGTs, GSTs) [45,46,47,48,49,50], which could contribute to the observed increases in D-glucaric acid and mercapturic acids. However, given the relatively small magnitude of weight and BMI changes in our cohort, it is unlikely that these modest anthropometric alterations fully account for the observed effects on hepatic detoxification enzymes. Moreover, because no significant group or group × time interactions were detected for weight or BMI, the differential pattern of detoxification capacity does not appear to be driven by these anthropometric changes.

The observed correlations between changes in metabolic and anthropometric variables suggest coordinated physiological responses to the intervention. The positive associations of D-glucaric acid and mercapturic acids with body weight and BMI imply that enhanced detoxification activity may parallel metabolic shifts linked to body composition changes. In contrast, the inverse relationship between D-glucaric acid and creatinine may reflect improved renal clearance or altered protein metabolism accompanying metabolic adaptation. These findings highlight an interconnected response between xenobiotic metabolism and systemic physiological changes, suggesting that metabolic detoxification pathways could serve as sensitive indicators of intervention-induced alterations in body composition and metabolic health.

Overall, the findings of the present study have several important implications. Measuring both D-glucaric acid and mercapturic acids provides complementary insight into detoxification, as parallel increases indicate responsiveness of glucuronide-related and GSH-mercapturate pathways. The sex difference in D-glucaric acid suggests that sex or hormone status should be considered in study design and interpretation. Since detoxification capacity did not moderate changes, refinement is needed, potentially incorporating functional enzyme assays (UGT, GST), genetic polymorphisms, gut microbiome analyses, and glutathione status. Future studies should more precisely measure exposure sources and control for urinary dilution, ideally with multiple time points to capture biomarker kinetics. The modest weight and BMI reductions observed alongside biomarker changes suggest that integrating lifestyle interventions could clarify how metabolic health interacts with detoxification, for example, by examining whether improvements in metabolic health correspond to increases in D-glucaric acid and mercapturic acids.

This study has several important strengths alongside notable limitations. Key strengths include the simultaneous measurement of two complementary Phase II detoxification biomarkers (D-glucaric acid and mercapturic acids), providing insight into distinct detoxification pathways. This study is the first of its kind to assess these biomarkers in relation to genetically defined detoxification capacity within a real-world cohort, offering novel insights into interindividual variation in detoxification biology. The inclusion of genetics-based stratification allowed explicit testing of genotypic moderation hypotheses, and although this classification did not explain the biomarker response, it represents an important step toward evaluating inherent detoxification capacity. However, several limitations constrain the interpretation of results. A key limitation of this study is the lack of a non-supplemented control group, which limits causal inference. As such, the observed biomarker changes should be considered preliminary and interpreted as potential effects pending confirmation in randomized controlled trials. The focus on a single occupational group (consisting of generally healthy individuals), was intended to minimize heterogeneity in environmental exposures and lifestyle factors but may limit the generalizability of the findings to more diverse populations with varying demographic and occupational backgrounds. The overall sample size (*n* = 30), and particularly the small subgroup sizes (LDC *n* = 18; NDC *n* = 12), reduce statistical power to detect moderate genotype × time interactions or sex-specific effects. The primary aim of the present pilot study was to assess feasibility and explore preliminary biomarker trends, which highlight the need for future, larger-scale studies to validate and extend these findings. In addition, lifestyle changes during the study period may have acted as minor confounding factors, influencing detoxification biomarkers independently of genetic capacity; therefore, additional uncontrolled dietary or behavioral factors may have also contributed to variability. Additional limitations include the lack of compound-specific mercapturate speciation (e.g., S-PMA, AAMA, GAMA, and SBMA), which constrains source attribution. The predefined NDC/LDC classification targeted several canonical detox genes (*CYP1A1*, *GSTM1*, *GSTT1*, *GSTP1*, and *NQO1*) but provides only a partial representation of overall detoxification capacity and did not include UGT isoforms (*UGT1A* and *UGT2B*) or direct measures of β-glucuronidase activity. Furthermore, although several models showed significant effects, deviations from normality suggest that the results should be interpreted with caution and verified in larger samples. Future studies would benefit from more detailed exposure assessment, spot-urine specific gravity or timed urine collections, and reporting of both normalized and unnormalized data.

## 4. Materials and Methods

### 4.1. Study Design and Participants

This was a pilot, prospective, open-label cohort study with pre- and post-intervention assessments conducted over an 8-week intervention period. The intervention consisted of a standardized dietary-supplement regimen, without investigational drugs or clinical pharmacologic procedures. The study was conducted within a single occupational cohort (hotel workforce) in Austria to minimize heterogeneity in environmental exposures. No randomization or placebo control was implemented, as the primary objective was to explore feasibility and biomarker responsiveness under real-world conditions rather than to test efficacy in a controlled trial setting. Participants were recruited from employees of a single hotel in Austria. Recruitment was completed via workplace notices, informational sessions delivered by study staff, and direct invitations to eligible employees. Recruitment materials emphasized voluntariness and clarified that refusal or withdrawal carried no professional consequences. All participants provided written informed consent prior to enrollment. The study was conducted in accordance with the Declaration of Helsinki and internationally recognized Good Clinical Practice standards. Data handling complied with the European Union General Data Protection Regulation and Austrian national regulations; all genetic and biochemical data were anonymized and stored on encrypted systems with role-based access. The study protocol was reviewed and approved by the DHGS Ethics Committee (ethical code: DHGS-EK-2025-005, 8 September 2025).

### 4.2. Inclusion and Exclusion Criteria

Eligible participants were adults aged 18 years or older employed within the participating hotel workforce, without a prior diagnosis of chronic liver disease or active hepatic dysfunction, who expressed willingness to undergo genetic testing and to comply with the 8-week supplementation regimen, and who were not taking medications known to substantially alter hepatic detoxification (e.g., chronic anticonvulsants or high-dose corticosteroids) at screening. Exclusion criteria included a known or diagnosed liver disease or clinically significant hepatic impairment, pregnancy or lactation, any known allergy or contraindication to supplement ingredients, highly inconsistent dietary or lifestyle patterns likely to confound biomarker analysis (such as planned prolonged travel or major dietary changes), and lack of informed consent to participate.

### 4.3. Procedures and Group Allocation

After obtaining informed consent, baseline assessments at Week 0 (second week of January 2025) included: medical history and lifestyle questionnaire, height and weight (and calculated BMI), collection of a dried blood spot for genotyping, and first-morning spot urine for biomarker measurement. Then, participants received study supplements and instructions.

Participants were stratified into two analytic groups based on genotyping results. The Limited Genetic Detox Capacity (LDC) group comprised individuals carrying deleterious polymorphisms or null variants in two or more target detoxification genes, including *GSTM1* null, *GSTT1* null, or functional variants in *GSTP1*, *NQO1*, or *CYP1A1* [8,48]. The Normal Genetic Detox Capacity (NDC) group included participants without high-risk genotypes in the specified detoxification genes [10,51], serving as the reference group for comparative analyses.

### 4.4. Intervention and Protocol Compliance

Participants received a standardized supplement package for 8 weeks. The products included Liv Ultra Leber Komplex, R-Alpha Lipoic Acid (bio-enhanced), Liposomal Glutathione, PMA-Zeolith Powder, N-Acetyl-Cysteine (NAC), and DIM (broccoli) extract, all manufactured by Sunday Natural GmbH (Berlin, Germany). Dosing followed the manufacturer’s label recommendations, and the exact lot numbers, batch information, and declared label doses were recorded for each participant and stored in the trial master file to ensure traceability. This supplement combination was selected to target complementary liver detoxification pathways, including Phase I and II biotransformation, antioxidant support, and modulation of glucuronidation and glutathione-dependent conjugation. Any participant-level dose modifications for tolerability were documented. Participants were instructed to maintain their habitual diet and lifestyle throughout the study and to avoid initiating new medications or supplements known to affect hepatic detoxification during the 8-week period. Table 8 provides a summary of individual supplement doses, proposed mechanisms, and supporting references.

Adherence was assessed using several complementary approaches, including participant-maintained daily online logs, a scheduled mid-study check-in at Week 4 (second week of February 2025) during which adherence logs were reviewed and a brief structured interview addressed potential adverse events and compliance, and the return and count of unused product units at Week 8. Participants were instructed to contact the study team immediately for any concerning symptoms. Adherence was calculated as the proportion of doses taken relative to the total prescribed over the 8-week intervention, with participants achieving an average adherence of ≥80% classified as meeting the protocol threshold for inclusion in primary analyses. In addition, an intention-to-treat analysis that included all enrolled participants was performed, and patterns of non-adherence were systematically recorded and evaluated as potential covariates in secondary analyses.

### 4.5. Biospecimen Collection, Processing, and Laboratory Analysis

First-morning spot urine samples were collected at Week 0 and Week 8 (January–March 2025). Participants provided midstream first-void samples into sterile containers, which were transported to the laboratory on ice within 4 h. Upon arrival, samples were centrifuged to remove particulate matter, aliquoted into a minimum of two 1–2 mL portions and stored at −80 °C until batch analysis. Urinary creatinine was measured independently using enzymatic methods to assess renal excretory consistency and support interpretation of metabolite concentrations. Concentrations of D-glucaric acid and mercapturic acids were normalized to urinary creatinine (expressed as nmol/mg creatinine for D-glucaric acid and μmol/mmol creatinine for mercapturic acids) to control for variability in hydration and renal output.

D-glucaric acid was quantified by high-performance liquid chromatography with UV detection (HPLC-UV) following validated sample preparation steps including centrifugation, deproteinization, dilution, and filtration [64,65]. Calibration curves from certified standards were included in each batch, along with low-/medium-/high-quality controls and sample duplicates; intra-assay CV ≤ 10% and inter-assay CV ≤ 15% were required. Mercapturic acids were measured using a validated enzymatic or kit-based assay (which included quality controls to ensure precision and accuracy), with optional LC-MS/MS confirmation for selected samples [15,16]. Collectively, this established analytical workflow provided validated performance characteristics, including standard calibration, quality-controlled precision, and robust sample preparation, ensuring the reliability of the reported biomarker data.

Finger-prick blood samples were collected at Week 0 onto certified filter paper cards, dried at ambient temperature for at least 3 h, packaged with desiccant, and stored at −20 °C until DNA extraction. DNA was extracted using validated silica-membrane or magnetic bead protocols, quantified spectrophotometrically or fluorometrically, and stored at −80 °C until genotyping. Genotyping targeted functional polymorphisms in Phase I/II detoxification genes (*CYP1A1*, *GSTM1*, *GSTP1*, *GSTT1*, *NQO1*) and histamine metabolism genes (*HNMT*, *DAO*). Single-nucleotide variants were detected using real-time PCR with allele-specific probes, while *GSTM1* and *GSTT1* null variants were identified via multiplex PCR with internal amplification controls [66,67]. Quality control included no-template and positive controls, with duplicate genotyping of ≥10% of samples; acceptable thresholds were ≥98% genotype call rate and ≥99% duplicate concordance.

### 4.6. Outcome Measures

Urinary D-glucaric acid and total mercapturic acids concentrations (along with urinary creatinine levels) were measured at baseline (Week 0) and post-intervention (Week 8) as continuous variables. Both biomarkers were also classified into five interpretive categories: low, marginally low, normal, marginally high, and high, based on predefined reference thresholds. Body weight was recorded in kilograms using calibrated digital scales with participants in light clothing and without shoes. Height was measured in centimeters using a wall-mounted stadiometer. Body Mass Index (BMI) was calculated as weight in kilograms divided by height in meters squared (kg/m^2^). Anthropometric measures were assessed at baseline and Week 8. Figure 3 presents a schematic overview of the study design and procedural workflow.

### 4.7. Statistical Analysis

All statistical analyses were conducted in R (version 4.5.1) using established packages for mixed modeling, ordinal regression, and visualization. Continuous variables are presented as mean ± SD and categorical variables as counts and percentages. Between-group differences at baseline were assessed using Pearson chi-square tests for nominal variables and Kruskal–Wallis tests for ordinal and continuous variables. Raincloud plots were generated (using *ggplot2* and *ggrain* packages) to visualize the distributional characteristics of key variables across groups and timepoints, combining raw data points, density plots, and boxplots in a single layered display. Linear mixed-effects models were conducted to assess within- and between-group changes over time. Fixed effects included time (pre vs. post), group (NDC vs. LDC), sex (male vs. female), and the time × group interaction, while random effects comprised participant-specific intercepts to account for repeated measures. Models were estimated with restricted maximum likelihood, and effect estimates were reported as regression coefficients (β), 95% confidence intervals, and p-values. Model fit and assumptions were evaluated by calculating marginal and conditional R^2^, inspecting residual versus fitted plots for homoscedasticity, and testing residual normality using the Shapiro–Wilk test. Missing data were handled by casewise exclusion (complete-case analysis), as the proportion of missing values was minimal and did not exhibit systematic patterns across groups or timepoints. Changes in biomarker categories were analyzed using two-sided Fisher’s exact tests. Ordinal logistic regression, adjusted for sex, estimated odds ratios for shifts in biomarker categories. Pairwise correlations among changes (Δ, post–pre intervention) in D-glucaric acid, mercapturic acids, body weight, BMI, and creatinine were assessed using Pearson’s correlation coefficients, and the results were visualized as both a correlation network and heatmap. Statistical significance was set at *p* < 0.05.

## 5. Conclusions

This study demonstrates that an 8-week liver support supplementation regimen was associated with coherent and biologically plausible increases in both D-glucaric acid and mercapturic acid levels, along with modest reductions in body weight and BMI. These findings may reflect enhanced glucuronide turnover and glutathione conjugation activity, or increased availability of related substrates, suggesting that detoxification pathways can exhibit measurable changes even among individuals with limited or normal genetic capacity. The lack of effect of genotype-based detoxification classification suggests that refining stratification to include UGT variation and microbiome β-glucuronidase may improve predictive value, while also emphasizing that exposure and physiological state often drive biomarker variability in real-world cohorts. Future studies should incorporate functional enzyme assays, gut microbiome profiling, repeated timed urine collections, and targeted supplementation trials to elucidate mechanisms, establish causality, and determine whether interventions can safely and durably enhance human detoxification with potential health benefits.

Overall, the present study may help inform the development of personalized nutritional strategies and targeted supplementation protocols aimed at optimizing detoxification capacity in everyday settings. More broadly, it may provide a translational foundation for integrating detoxification biomarkers into individualized health monitoring and preventive care.

## Figures and Tables

**Figure 1 ijms-26-10209-f001:**
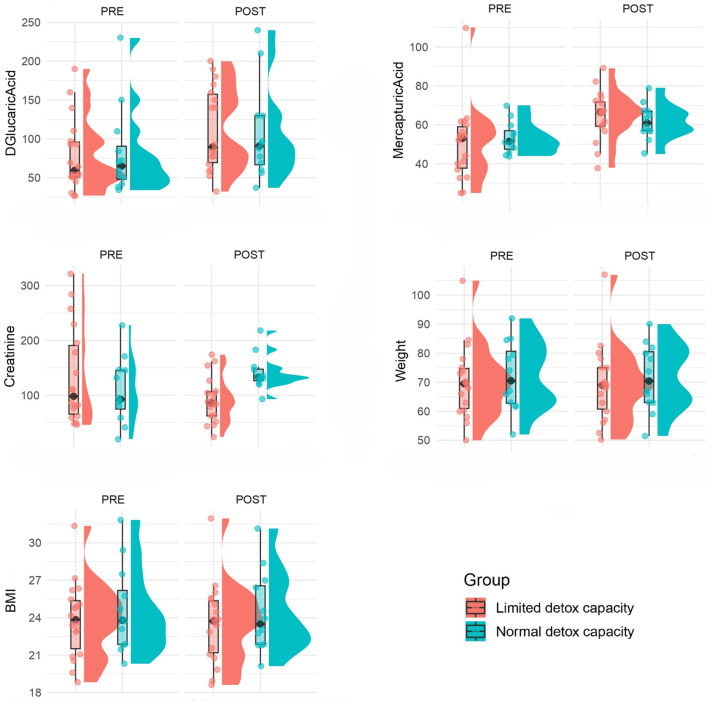
Raincloud plots showing biochemical and physiological outcomes before (PRE) and after (POST) the liver support supplementation intervention. Measures include D-glucaric acid, mercapturic acids, creatinine, weight, and body mass index (BMI). D-glucaric acid and mercapturic acid values were normalized to urinary creatinine (nmol/mg creatinine for D-glucaric acid; μmol/mmol creatinine for mercapturic acids). Data are stratified by detoxification capacity (limited detox capacity: *n* = 18, red; normal detox capacity: *n* = 12, blue). Each plot displays the distribution (half-violin), individual data points, individual data points, and summary statistics (boxplot showing the median and interquartile range, with whiskers extending to the minimum and maximum values).

**Figure 2 ijms-26-10209-f002:**
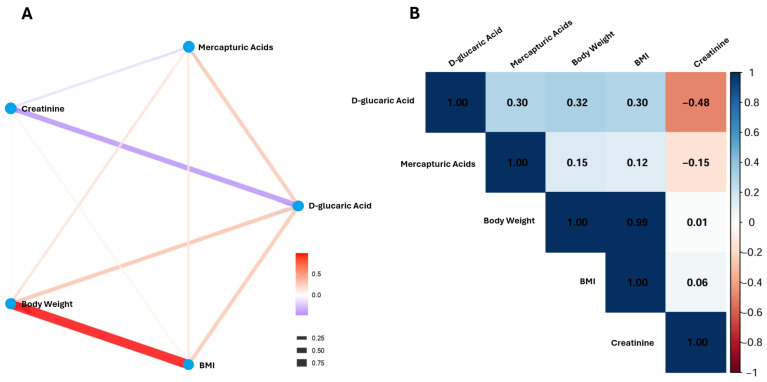
Correlation patterns among changes in metabolic and anthropometric variables following the intervention. (**A**) Correlation network illustrating the interrelationships between the changes (Δ, post–pre) in the study variables. Each node represents a variable, and the connecting edges denote pairwise correlations between their changes. The color of each edge indicates the direction of the correlation (red = positive, purple = negative), while edge thickness reflects the strength of the association. (**B**) Correlation heatmap displaying the pairwise correlation coefficients among the same variables. Color intensity corresponds to the strength and direction of correlation (blue shades = negative; red shades = positive), as indicated by the color scale bar. Numerical values within each cell represent the correlation coefficients (*r*).

**Figure 3 ijms-26-10209-f003:**
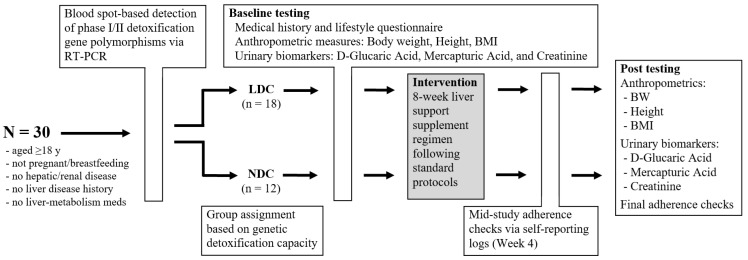
Schematic of study design. RT-PCR: real-time polymerase chain reaction; BMI: body mass index; LDC: limited detox capacity; NDC: normal detox capacity.

**Table 1 ijms-26-10209-t001:** Baseline characteristics of participants by study groups (LDC vs. NDC).

		Total (*n* = 30)	LDC (*n* = 18)	NDC (*n* = 12)	Statistical Difference
Age (years)		35.77 ± 10.88	33.94 ± 9.65	38.50 ± 12.44	*p* = 0.4
Sex	Female	20 (67%)	13 (72%)	7 (58%)	*p* = 0.5
	Male	10 (33%)	5 (28%)	5 (42%)	
Body Weight (kg)		70.91 ± 12.21	70.17 ± 12.65	72.02 ± 11.96	*p* = 0.6
Height (cm)		1.71 ± 0.08	1.71 ± 0.08	1.71 ± 0.07	*p* = 0.9
BMI (kg/m^2^)		24.15 ± 3.19	23.84 ± 3.05	24.61 ± 3.47	*p* = 0.7
D-Glucaric Acid (nmol/mg)		80.03 ± 50.16	78.22 ± 46.69	82.75 ± 57.02	*p* = 0.9
D-Glucaric Acid Levels	Low	4 (13%)	2 (11%)	2 (17%)	*p* = 0.7
	Marginally low	13 (43%)	9 (50%)	4 (33%)	
	Normal	13 (43%)	7 (39%)	6 (50%)	
Mercapturic Acids (µmol/mmol)		51.53 ± 15.83	50.28 ± 19.49	53.42 ± 8.22	*p* = 0.4
Mercapturic Acid Levels	Low	6 (20%)	6 (33%)	0 (0%)	*p* = 0.8
	Marginally low	5 (17%)	2 (11%)	3 (25%)	
	Normal	17 (57%)	9 (50%)	8 (67%)	
	Marginally high	1 (3.3%)	0 (0%)	1 (8.3%)	
	High	1 (3.3%)	1 (5.6%)	0 (0%)	
Creatinine (mg/dL)		123.73 ± 78.02	134.28 ± 88.57	107.92 ± 58.92	*p* = 0.7

LDC: limited detox capacity; NDC: normal detox capacity; BMI: body mass index.

**Table 2 ijms-26-10209-t002:** Results of linear mixed-effects model for body weight with fixed effects estimates.

	Estimate (β)	95% CI	SE	DF	Statistic	*p*-Value
Intercept	66.73	[61.12, 72.34]	2.75	30.3	t = 24.27	<0.001
Time (post vs. pre)	−0.87	[−1.60, −0.15]	0.35	30	t = −2.46	0.020
Group (NDC vs. LDC)	0.13	[−8.02, 8.28]	3.99	30.3	t = 0.03	0.974
Sex (male vs. female)	12.37	[3.92, 20.82]	4.14	30	t = 2.99	0.006
Time × Group	0.30	[−0.85, 1.44]	0.56	30	t = 0.53	0.600

Note: Model fit indices: marginal R^2^ (variance explained by fixed effects) = 0.24, conditional R^2^ (variance explained including random effects) = 0.99. Residuals were normally distributed (Shapiro–Wilk W = 0.97, *p* = 0.22). CI: Confidence Interval; SE: Standard Error; DF: Degrees of Freedom; LDC: Limited Detox Capacity; NDC: Normal Detox Capacity.

**Table 3 ijms-26-10209-t003:** Results of linear mixed-effects model for BMI with fixed effects estimates.

	**Estimate (β)**	**95% CI**	**SE**	**DF**	**Statistic**	***p*-Value**
Intercept	23.62	[21.99, 25.26]	0.80	30.4	t = 29.48	<0.001
Time (post vs. pre)	−0.31	[−0.56, −0.06]	0.12	30	t = −2.49	0.019
Group (NDC vs. LDC)	0.66	[−1.71, 3.04]	1.16	30.4	t = 0.57	0.572
Sex (male vs. female)	0.77	[−1.69, 3.23]	1.20	30	t = 0.64	0.528
Time × Group	0.10	[−0.30, 0.50]	0.20	30	t = 0.52	0.610

Note: Model fit indices: marginal R^2^ (variance explained by fixed effects) = 0.03, conditional R^2^ (variance explained including random effects) = 0.99. Residuals were normally distributed (Shapiro–Wilk W = 0.98, *p* = 0.43). CI: Confidence Interval; SE: Standard Error; DF: Degrees of Freedom; LDC: Limited Detox Capacity; NDC: Normal Detox Capacity.

**Table 4 ijms-26-10209-t004:** Results of linear mixed-effects model for D-glucaric Acid with fixed effects estimates.

	Estimate (β)	95% CI	SE	DF	Statistic	*p*-Value
Intercept	90.22	[65.86, 114.57]	12.14	51.6	t = 7.43	<0.001
Time (post vs. pre)	33.83	[6.73, 60.93]	13.27	30	t = 2.55	0.016
Group (NDC vs. LDC)	10.53	[−25.75, 46.80]	18.10	54.2	t = 0.58	0.563
Sex (male vs. female)	−43.19	[−74.48, −11.89]	15.32	30	t = −2.82	0.008
Time × Group	−9.25	[−52.10, 33.60]	20.98	30	t = −0.44	0.662

Note: Model fit indices: marginal R^2^ (variance explained by fixed effects) = 0.22, conditional R^2^ (variance explained including random effects) = 0.47. Residuals deviated from normality (Shapiro–Wilk W = 0.95, *p* = 0.01). CI: Confidence Interval; SE: Standard Error; DF: Degrees of Freedom; LDC: Limited Detox Capacity; NDC: Normal Detox Capacity.

**Table 5 ijms-26-10209-t005:** Results of linear mixed-effects model for Mercapturic Acids with fixed effects estimates.

	Estimate (β)	95% CI	SE	DF	Statistic	*p*-Value
Intercept	48.88	[42.32, 55.45]	3.28	58.41	t = 14.91	<0.001
Time (post vs. pre)	14.67	[6.02, 23.32]	4.24	30	t = 3.46	0.002
Group (NDC vs. LDC)	2.44	[−7.44, 12.32]	4.94	59.61	t = 0.49	0.623
Sex (male vs. female)	5.01	[−2.69, 12.72]	3.77	30	t = 1.33	0.194
Time × Group	−6.42	[−20.09, 7.26]	6.70	30	t = −0.96	0.346

Note: Model fit indices: marginal R^2^ (variance explained by fixed effects) = 0.21, conditional R^2^ (variance explained including random effects) = 0.26. Residuals deviated from normality (Shapiro–Wilk W = 0.92, *p* = 0.001). CI: Confidence Interval; SE: Standard Error; DF: Degrees of Freedom; LDC: Limited Detox Capacity; NDC: Normal Detox Capacity.

**Table 6 ijms-26-10209-t006:** Results of linear mixed-effects model for Creatinine with fixed effects estimates.

	Estimate (β)	95% CI	SE	DF	Statistic	*p*-Value
Intercept	126.62	[42.32, 55.45]	14.42	60	t = 8.78	<0.001
Time (post vs. pre)	−42.89	[6.02, 23.32]	19.39	60	t = −2.21	0.031
Group (NDC vs. LDC)	−30.19	[−7.44, 12.32]	21.79	60	t = −1.39	0.171
Sex (male vs. female)	27.56	[−2.69, 12.72]	16.10	60	t = 1.71	0.092
Time × Group	76.22	[−20.09, 7.26]	30.65	60	t = 2.49	0.016

Note: Model fit indices: marginal R^2^ (variance explained by fixed effects) = 0.15, conditional R^2^ (variance explained including random effects) = 0.15. Residuals deviated from normality (Shapiro–Wilk W = 0.94, *p* = 0.003). CI: Confidence Interval; SE: Standard Error; DF: Degrees of Freedom; LDC: Limited Detox Capacity; NDC: Normal Detox Capacity.

**Table 7 ijms-26-10209-t007:** Between-group comparison of changes in detoxification biomarker levels and overall response following the intervention.

		LDC (*n* = 18)	NDC (*n* = 12)	Between-Group Difference ^a^	Adjusted Odds Ratio ^b^
D-Glucaric Acid Level Changes	Improved	10 (56%)	5 (42%)	*p* = 0.924	2.16 (*p* = 0.310)
	Remained marginally low	1 (6%)	1 (8%)		
	Remained normal	6 (33%)	5 (42%)		
	Worsened slightly	1 (6%)	1 (8%)		
Mercapturic Acid Level Changes	Improved	13 (72%)	5 (42%)	*p* = 0.165	3.33 (*p* = 0.124)
	Returned to normal	1 (6%)	1 (8%)		
	Remained normal	4 (22%)	6 (50%)		
Overall Response to Intervention	Dual improvement	7 (39%)	1 (8%)	*p* = 0.263	2.88 (*p* = 0.150)
	Single improvement	7 (39%)	8 (67%)		
	Stable	2 (11%)	2 (17%)		
	Mixed responses	2 (11%)	1 (8%)		

^a^ Two-sided Fisher’s exact test comparing LDC (limited detox capacity) vs. NDC (normal detox capacity) groups. ^b^ Odds ratio and corresponding *p*-value from ordinal logistic regression for LDC vs. NDC, adjusted for sex.

**Table 8 ijms-26-10209-t008:** Supplement details and mechanistic rationale.

Supplement	Daily Dosage	Mechanistic Role and Synergistic Potential	References
Liposomal Glutathione	1 capsule	Acts as a direct antioxidant and key cofactor for Phase II conjugation reactions, facilitating detoxification of reactive species. Enhances overall detoxification capacity and regenerates α-lipoic acid; synergistic with NAC for maintaining intracellular glutathione levels.	[52,53]
N-Acetylcysteine (NAC)	1 capsule	Serves as a cysteine donor and precursor for glutathione synthesis, supporting conjugation of xenobiotics and antioxidant defense. Works synergistically with glutathione and α-lipoic acid to sustain redox balance and detoxification efficiency.	[54,55]
R-Alpha Lipoic Acid	2 capsules	A potent antioxidant that regenerates glutathione and vitamins C and E while supporting mitochondrial energy metabolism. Enhances the intracellular antioxidant network and potentiates the effects of NAC and glutathione.	[56,57]
DIM (Diindolylmethane, broccoli extract)	2 capsules	Modulates Phase I and Phase II detoxification enzymes, promotes balanced estrogen metabolism, and provides antioxidant protection. May enhance glucuronidation and sulfation pathways and act synergistically with NAC and glutathione in detox support.	[58,59]
PMA-Zeolith Powder (Zeolite)	2 × 3 g	Binds heavy metals and toxins in the gastrointestinal tract, reducing enterohepatic recirculation and systemic toxin load. Complements glutathione- and NAC-dependent detoxification pathways for toxin elimination.	[60,61]
Liv Ultra Leber Komplex	3 capsules	Multi-ingredient liver support formulation containing hepatoprotective herbs, choline, vitamins, and antioxidants that enhance Phase II detoxification and hepatic resilience. Acts synergistically with other supplements to strengthen overall liver and metabolic detox pathways.	[62,63]

## Data Availability

The original contributions presented in this study are included in the article. Further inquiries can be directed to the corresponding authors.

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
