# Peer review of "Does Genetic Variation in Detoxification Capacity Influence Hepatic Biomarker Responses to a Liver Support Supplementation Regimen?"

_ijms, 2025, doi:10.3390/ijms262010209_

Round 1

Reviewer 1 Report

Comments and Suggestions for Authors

This well-conceived and methodologically coherent pilot study investigates the interaction between genetic variability in detoxification capacity and biochemical responses to a multi-ingredient liver support supplementation regimen. This is a timely topic of growing scientific and clinical relevance as it bridges nutrigenomics, hepatology and functional nutrition.

I have made some suggestions on how you could improve your work. This doesn't mean that you have to agree with them or rewrite your work in the same way. They are just suggestions to help you see things from a different perspective.

1) In the 'Introduction' section, please, add 1-2 sentences after line 49 explaining why enhancing detoxification capacity matters beyond biochemical curiosity, for example, in relation to environmental toxin burden, metabolic health or the prevention of chronic diseases.

2) Please, indicate that, although many commercial 'liver support' supplements are available, scientific validation remains limited.

3) If your cohort includes both men and women or people of different ages, you could mention that detoxification enzyme activity can vary according to sex, hormonal status or age.

4) While the final paragraph outlines the aims of the study, you could emphasise what distinguishes your study from previous work, for example the simultaneous integration of genotyping, biomarker tracking and adherence-controlled supplementation in a real-world setting.

5) In the 'Materials and Methods' section, the decision to focus on a single occupational group is explained as a means of minimising heterogeneity. However, you could briefly discuss any potential limitations or concerns regarding generalisability.

6) Although pilot studies often omit formal power analyses, it is useful to mention whether the sample size was chosen pragmatically or based on prior variability estimates.

7) You list product names and general ingredients, but you could also include a table summarising active compounds, dosage per serving and manufacturer for reproducibility purposes.

8) While it is stated as an open-label cohort, a brief acknowledgement that no randomisation or placebo control was implemented (and an explanation of why this was the case) would improve transparency.

9) The statistical analysis section should explain how missing values were handled (exclusion, imputation or model-based approaches).

10) Since urinary biomarkers can vary with hydration, please specify whether the concentrations of D-glucaric acid and mercapturic acid were expressed relative to creatinine (µmol/mmol creatinine) or as raw concentrations.

11) You provided QC limits for HPLC-UV, but the mercapturic acid assay lacks similar performance metrics.

12) You specify R for the analysis, but you could also note which packages were used for plotting.

13) Creatinine measurement appears twice (lines 397-398 and 422-423). Keep one detailed mention (preferably in section 4.5) and refer to it briefly in section 4.6.

14) Replace 'Mercapturic acids were measured' (line 404) with 'Mercapturic acids were measured'.

15) In the 'Results' section, please, provide a concise overview of the main findings before delving into the numerical details. For example, you could write: 'Overall, the intervention was associated with modest but significant increases in Phase II detoxification biomarkers (D-glucaric acid and mercapturic acids), without major group differences between those with limited and normal detoxification capacity.' This helps readers to grasp the direction of the findings before interpreting the detailed tables.

16) Currently, each model is presented in a near-identical format, which is rigorous but somewhat repetitive. Please, add short interpretive transitions before or after each main result (e.g. 'This finding suggests that...'), for instance: After Table 4 (D-glucaric acid): 'This result indicates an overall upregulation of glucuronidation-related activity over the intervention period, independent of detox capacity group.' After Table 5 (mercapturic acids): 'The increase in mercapturic acids suggests enhanced GSH-dependent detoxification following supplementation.'

17) Since several models violated normality assumptions, include a brief acknowledgement: 'Although several models showed significant effects, deviations from normality suggest that the results should be interpreted with caution and verified in larger samples.'

18) Beyond p-values, indicate whether changes are clinically or biologically meaningful.

19) If the data permit, please, add exploratory correlations (e.g. between D-glucaric acid and mercapturic acids) to illustrate coherence between detoxification pathways.

20) There is a numbering error: the final table is labelled “Table 5” again for creatinine results, but it should be Table 6. Ensure that all references to tables and figures match the actual numbering.

21) 'Satble' in Table 7 should be corrected to 'stable'.

22) In the 'Discussion' section, after presenting biochemical and mechanistic explanations, please, add a summary paragraph linking the findings to their broader physiological relevance. This will provide readers with a 'big-picture' overview rather than leaving the discussion segmented by variable type.

23) The discussion currently notes 'modest but statistically significant increases', but does not address the clinical or functional significance of these increases. Please, add information on the clinical or public health significance of enhanced detoxification. This will help avoid the impression that the findings are statistically significant but biologically meaningless.

24) The genotype discussion (lines 224-251) is valuable, but dense and repetitive. Consider condensing the points that overlap about GST variability and focusing more on future research directions, such as multi-gene panels and gene–environment interactions and the interplay between the microbiome and genotype.

25) Lines 270-285 - You reference relevant studies, but you could make the comparison more explicit - for example, 'our findings align with/contrast with previous human trials'.

26) Please, address external validity and limitations in the study design (lines 309-327). You mention sample size and the lack of compound-specific analysis, but you could also briefly acknowledge the following: i) limited generalisability due to the small, likely healthy cohort; ii) uncontrolled dietary or lifestyle confounders beyond weight and BMI.

27) Lines 231-233 and 243-245 repeat the idea that 'linking polymorphisms to detox outcomes is complex'. Keep one well-articulated version.

28) Lines 320-322 could be shortened to 'Lifestyle changes during the study period may have acted as minor confounding factors, influencing detoxification biomarkers independently of genetic capacity.'

29) Your 'Conclusions' section is clear and logically connected to the 'Discussion', providing a summary of the findings. However, it could conclude with a more forward-looking and integrative statement emphasising the broader implications (e.g. for nutritional science, toxicology or personalised health).

While this manuscript presents valuable findings, improving the clarity and depth of the discussion and the level of methodological detail would further enhance its impact. The findings are well supported by the experimental approaches employed. I recommend accepting the manuscript with some revisions.

Author Response

Please find attached- thank you!

Dear Reviewer #1,
Dear Editors,

Thank you for your time, constructive feedback, and the opportunity to revise the manuscript ijms-3934416, entitled “Does Genetic Variation in Detoxification Capacity Influence Hepatic Biomarker Responses to a Liver Support Supplementation Regimen?

In response to your valuable comments, we have provided supporting evidence along with detailed explanations that directly address each point. We hope our responses below, along with the revisions throughout the manuscript, adequately address your concerns. All changes throughout the manuscript have been highlighted in yellow. Additionally, the reference list has been updated based on the revisions, with new references highlighted in yellow.

Kind Regards,
Corresponding authors, on behalf of all authors

Review report 1

This well-conceived and methodologically coherent pilot study investigates the interaction between genetic variability in detoxification capacity and biochemical responses to a multi-ingredient liver support supplementation regimen. This is a timely topic of growing scientific and clinical relevance as it bridges nutrigenomics, hepatology and functional nutrition.

I have made some suggestions on how you could improve your work. This doesn't mean that you have to agree with them or rewrite your work in the same way. They are just suggestions to help you see things from a different perspective.

Response: We sincerely appreciate the time you have taken to review our manuscript and for sharing such positive feedback. This encouragement strengthens our motivation to further improve the quality of our work. We are pleased to confirm that we have addressed almost all your suggestions, with the exception of a few points where we provided reasoned justifications instead of implementing changes. Thank you once more!

1) In the 'Introduction' section, please, add 1-2 sentences after line 49 explaining why enhancing detoxification capacity matters beyond biochemical curiosity, for example, in relation to environmental toxin burden, metabolic health or the prevention of chronic diseases.

Response: Thank you. We have added a statement to clarify the clinical relevance of enhancing hepatic detoxification (lines 50-52).

2) Please, indicate that, although many commercial 'liver support' supplements are available, scientific validation remains limited.

Response: Thanks for your comment. We have included this point accordingly (lines 85-87).

3) If your cohort includes both men and women or people of different ages, you could mention that detoxification enzyme activity can vary according to sex, hormonal status or age.

Response: Thank you very much. We have included this point accordingly (lines 57-58).

4) While the final paragraph outlines the aims of the study, you could emphasise what distinguishes your study from previous work, for example the simultaneous integration of genotyping, biomarker tracking and adherence-controlled supplementation in a real-world setting.

Response: Thank you for your comment. We agree and have addressed this point accordingly (lines 95-98).

5) In the 'Materials and Methods' section, the decision to focus on a single occupational group is explained as a means of minimising heterogeneity. However, you could briefly discuss any potential limitations or concerns regarding generalisability.

Response: Thank you for your valuable comment. We agree and have added a statement in the Limitations section noting that the focus on a single occupational group, while reducing heterogeneity, may limit the generalizability of the findings to broader populations (lines 387-391).

6) Although pilot studies often omit formal power analyses, it is useful to mention whether the sample size was chosen pragmatically or based on prior variability estimates.

Response: We appreciate the reviewer’s concern regarding statistical power. As a pilot study, our primary objective was to assess feasibility, explore biomarker responsiveness, and identify preliminary trends across genotype groups. The sample size was intentionally limited to allow for controlled implementation and close monitoring of adherence and biological responses. While we acknowledge that this pilot study may be underpowered to detect modest genotype × time or sex-specific effects, our findings offer biologically plausible signals that warrant further investigation in this promising area. We have now provided further explanation of this limitation in the Discussion (lines 391-395), clarifying that these results are intended to inform the design and powering of future, larger-scale studies (including ongoing research currently underway in our lab).

7) You list product names and general ingredients, but you could also include a table summarising active compounds, dosage per serving and manufacturer for reproducibility purposes.

Response: Thank you for this constructive comment. In response, we have designed and added a table summarizing the supplement composition, individual dosages, biological mechanisms and supporting references, along with providing a justification for the combination used in our study (lines 461-463 and 467-470).

8) While it is stated as an open-label cohort, a brief acknowledgement that no randomisation or placebo control was implemented (and an explanation of why this was the case) would improve transparency.

Response: Thank you. We agree and have added a statement clarifying that no randomization or placebo control was implemented (lines 416-418).

9) The statistical analysis section should explain how missing values were handled (exclusion, imputation or model-based approaches).

Response: We appreciate your comment. We agree and have added a statement in the Statistical Analysis section clarifying how missing data were handled (lines 544–546).

10) Since urinary biomarkers can vary with hydration, please specify whether the concentrations of D-glucaric acid and mercapturic acid were expressed relative to creatinine (µmol/mmol creatinine) or as raw concentrations.

Response: Thank you very much for this comment. D-glucaric acid and mercapturic acid values have already been normalized to urinary creatinine (nmol/mg creatinine for D-glucaric acid; μmol/mmol creatinine for mercapturic acids). The normalization procedure has now been described in the Methods section and added to the figure legend as well (lines 488-491 and 136-138).

11) You provided QC limits for HPLC-UV, but the mercapturic acid assay lacks similar performance metrics.

Response: Thank you. We agree and have revised the text to shortly specify the quality controls for the mercapturic acid assay (lines 497-498).

12) You specify R for the analysis, but you could also note which packages were used for plotting.

Response: Thank you. We have now specified the R packages used for plotting in the revised Statistical Analysis section (line 534).

13) Creatinine measurement appears twice (lines 397-398 and 422-423). Keep one detailed mention (preferably in section 4.5) and refer to it briefly in section 4.6.

Response: Thank you for this observation. We have removed the repetitive statement, and it is now briefly specified in the opening of Section 4.6, alongside the other two biomarkers (lines 515-516).

14) Replace 'Mercapturic acids were measured' (line 404) with 'Mercapturic acids were measured'.

Response: Great catch- thank you! We have corrected "was" to "were" (line 497).

15) In the 'Results' section, please, provide a concise overview of the main findings before delving into the numerical details. For example, you could write: 'Overall, the intervention was associated with modest but significant increases in Phase II detoxification biomarkers (D-glucaric acid and mercapturic acids), without major group differences between those with limited and normal detoxification capacity.' This helps readers to grasp the direction of the findings before interpreting the detailed tables.

Response: Thank you for this suggestion. In response, we have added a concise summary statement in the Results section (after reporting the baseline statistics) to provide a clear overview of the main findings before presenting the detailed numerical results for each outcome measure (lines 116-120).

16) Currently, each model is presented in a near-identical format, which is rigorous but somewhat repetitive. Please, add short interpretive transitions before or after each main result (e.g. 'This finding suggests that...'), for instance: After Table 4 (D-glucaric acid): 'This result indicates an overall upregulation of glucuronidation-related activity over the intervention period, independent of detox capacity group.' After Table 5 (mercapturic acids): 'The increase in mercapturic acids suggests enhanced GSH-dependent detoxification following supplementation.'

Response: We thank the reviewer for this comment and appreciate the intent to make the Results section more readable. However, we would like to note that, as you know, the Results section is typically intended to present the findings in a purely descriptive and objective manner, without interpretive commentary. To maintain adherence to standard scientific reporting conventions, we have therefore kept the Results section focused on the statistical outcomes and numerical data. Interpretive statements regarding the biological or mechanistic implications of these findings have been fully discussed in the Discussion section, where such contextualization is more appropriate and allows for a nuanced interpretation. Thank you for your kind understanding.

17) Since several models violated normality assumptions, include a brief acknowledgement: 'Although several models showed significant effects, deviations from normality suggest that the results should be interpreted with caution and verified in larger samples.'

Response: Thank you for this constructive comment. We agree that residuals for D-glucaric acid, mercapturic acids, and creatinine deviated from normality, and for transparency, these deviations have been explicitly noted in the Results and table footnotes. As you know, linear mixed-effects models are generally robust to moderate deviations from normality, particularly when random effects are included. These models also provide unbiased estimates of fixed effects even under mild non-normality. To address this comment, however, we have added a statement in the Discussion highlighting this limitation and advising cautious interpretation of the findings (lines 404-406).

18) Beyond p-values, indicate whether changes are clinically or biologically meaningful.

Response: We understand the reviewer’s concern. However, we would like to again ask for your understanding, as, to adhere to standard scientific reporting, we intended to keep the Results section focused on presenting findings objectively, emphasizing statistical outcomes rather than commenting on their clinical or biological relevance. Thank you again!

19) If the data permit, please, add exploratory correlations (e.g. between D-glucaric acid and mercapturic acids) to illustrate coherence between detoxification pathways.

Response: We thank the reviewer for this constructive suggestion. In response, we performed exploratory correlation analyses to examine potential relationships between the study variables. The results of these analyses are now reported both in the text and through visual illustrations (Figure 2) to provide a clear overview of the coherence between detoxification pathways. Additionally, we have updated the Statistical Analysis section to briefly describe the methods used for these exploratory correlations. Relevant interpretations of the observed associations have also been incorporated into the Discussion section (lines 214-232, 352-361, and 548-551).

20) There is a numbering error: the final table is labelled “Table 5” again for creatinine results, but it should be Table 6. Ensure that all references to tables and figures match the actual numbering.

Response: Thank you. We have revised this typo (line 197).

21) 'Satble' in Table 7 should be corrected to 'stable'.

Response: Thank you. We have revised this typo (Table 7).

22) In the 'Discussion' section, after presenting biochemical and mechanistic explanations, please, add a summary paragraph linking the findings to their broader physiological relevance. This will provide readers with a 'big-picture' overview rather than leaving the discussion segmented by variable type.

Response: Thank you very much for this constructive comment. We have added two summary statements in the Discussion (after mechanistic explanations) linking our findings to broader physiological relevance, as suggested (lines 292-298).

23) The discussion currently notes 'modest but statistically significant increases', but does not address the clinical or functional significance of these increases. Please, add information on the clinical or public health significance of enhanced detoxification. This will help avoid the impression that the findings are statistically significant but biologically meaningless.

Response: We appreciate this insightful comment. Accordingly, we have clarified that the observed increases may reflect improved hepatic clearance, which could hold clinical relevance for individuals with reduced detoxification efficiency (lines 236-238).

24) The genotype discussion (lines 224-251) is valuable, but dense and repetitive. Consider condensing the points that overlap about GST variability and focusing more on future research directions, such as multi-gene panels and gene–environment interactions and the interplay between the microbiome and genotype.

Response: Thank you for this thoughtful suggestion. We have streamlined the genotype discussion to reduce redundancy and added a forward-looking statement on multi-gene panels, gene–environment interactions, and microbiome–genotype interplay. Together with the previously added statement in response to Comment 22, these revisions enhance conceptual clarity (lines 269-271, 273-276, and 298-300).

25) Lines 270-285 - You reference relevant studies, but you could make the comparison more explicit - for example, 'our findings align with/contrast with previous human trials'.

Response: Thank you very much. We have revised the associated statement in order to make the comparison with prior human trials more explicit (lines 331-334).

26) Please, address external validity and limitations in the study design (lines 309-327). You mention sample size and the lack of compound-specific analysis, but you could also briefly acknowledge the following: i) limited generalisability due to the small, likely healthy cohort; ii) uncontrolled dietary or lifestyle confounders beyond weight and BMI.

Response: We appreciate the reviewer’s request. We have revised the associated paragraph to clearly address the abovementioned points (lines 387-398).

27) Lines 231-233 and 243-245 repeat the idea that 'linking polymorphisms to detox outcomes is complex'. Keep one well-articulated version.

Response: Thank you. We have revised the associated statements accordingly (lines 273-276).

28) Lines 320-322 could be shortened to 'Lifestyle changes during the study period may have acted as minor confounding factors, influencing detoxification biomarkers independently of genetic capacity.'

Response: Thank you so much. We have replaced the suggestive statement as recommended and revised the subsequent sentence to enhance clarity and coherence, explicitly addressing potential dietary and lifestyle confounders that may have influenced biomarker variability (lines 396-398).

29) Your 'Conclusions' section is clear and logically connected to the 'Discussion', providing a summary of the findings. However, it could conclude with a more forward-looking and integrative statement emphasising the broader implications (e.g. for nutritional science, toxicology or personalised health).

Response: Thank you for your thoughtful suggestion. We have incorporated the recommended addition at the end of the Conclusion section to highlight the broader relevance of our findings for personalized nutrition strategies and preventive health application (lines 568-572).

While this manuscript presents valuable findings, improving the clarity and depth of the discussion and the level of methodological detail would further enhance its impact. The findings are well supported by the experimental approaches employed. I recommend accepting the manuscript with some revisions.

Response: We appreciate the reviewer’s positive evaluation of the scope and significance of our work, and hope that the revisions and clarifications provided sufficiently address your concerns.

Thank you once more!

Reviewer 2 Report

Comments and Suggestions for Authors

Manuscript Title:

Does Genetic Variation in Detoxification Capacity Influence Hepatic Biomarker Responses to a Liver Support Supplementation Regimen?

The study included only 30 participants divided into two genetic groups (18 LDC and 12 NDC). This sample is underpowered to detect modest genotype × time interactions or sex-specific effects. The authors should perform and report a power analysis or at least acknowledge this limitation more explicitly in the Discussion. The binary grouping (LDC/NDC) is based on limited polymorphisms (CYP1A1, GSTM1, GSTT1, GSTP1, NQO1). This classification is too simplistic and may not truly reflect “detoxification capacity.” Suggest incorporating or discussing additional relevant genes (e.g., UGT1A, UGT2B, β-glucuronidase activity) and justifying why only these five genes were chosen.

The sex difference (higher in females) is interesting but insufficiently explored. The authors could expand the discussion on hormonal regulation of UGT enzymes, microbiome β-glucuronidase, or sex hormone–dependent metabolism to give biological depth.

The mixture of supplements (glutathione, NAC, α-lipoic acid, DIM, zeolite) is complex. The manuscript lacks justification for their combination, individual dosages, or potential interactions. Include a table detailing doses, proposed mechanisms, and references supporting synergy.

The total sample (n = 30; LDC = 18, NDC = 12) is too small to detect genotype × time interactions with adequate power. Please include a power analysis or explicitly acknowledge this limitation and its impact on negative findings.

The binary classification into “Limited Detox Capacity” and “Normal Detox Capacity” based on only five genes (CYP1A1, GSTM1, GSTT1, GSTP1, NQO1) oversimplifies the complex detoxification network. Provide scientific justification for these markers, and discuss the omission of other critical pathways (UGTs, β-glucuronidase, etc.).

Without a non-supplemented control, biomarker changes cannot be confidently attributed to the intervention. The authors must clearly discuss this as a major limitation and avoid causal wording in the Abstract and Discussion.

It is unclear whether D-glucaric acid and mercapturic acid values were normalized to urinary creatinine. This step is essential to control for hydration and renal output variability. Please confirm and describe the normalization procedure in the Methods and figure legends.

Several mixed-effects models show non-normal residuals (Shapiro-Wilk p < 0.05). Authors should verify results using non-parametric analyses or data transformation to ensure robustness.

The discussion equates increased urinary levels with improved detoxification capacity. This may instead represent elevated excretion or turnover of conjugates. Revise interpretation and add references differentiating up-regulation vs. clearance phenomena.

The finding that females exhibit higher D-glucaric acid is intriguing but underexplored. Expand on possible hormonal or microbiome-related mechanisms (e.g., UGT2B17 expression, β-glucuronidase activity).

Weight and BMI decreased significantly; weight loss can independently affect hepatic enzyme activity. Consider performing an adjusted model or at least discuss this confounder.

The supplement mixture (glutathione, NAC, α-lipoic acid, DIM, zeolite) requires clearer rationale. Include a table summarizing ingredients, doses, proposed mechanisms, and supporting references.

The statement that supplementation “beneficially modulates detoxification regardless of genotype” overreaches the evidence. Rephrase to reflect that effects are potential and preliminary pending confirmation in randomized controlled trials.

Figures – Raincloud plots are elegant but busy; consider adding summary bar graphs (mean ± SEM).

Terminology – Ensure consistent italicization of gene names (CYP1A1, GSTM1, etc.).

Ethical Statement – Already adequate but add IRB approval date explicitly if available.

Clarity of Hypothesis – Restate hypothesis explicitly at the end of the Introduction, indicating expected biomarker changes.

  • Add “pilot open-label study” to clarify scope.

Specify normalization approach (to creatinine or not).

  • Soften conclusions (replace “beneficially modulates” with “was associated with changes in”).
  • Excellent ethical and biospecimen handling details.
  • Add exact supplement dosages and average adherence rate.
  • Clarify analytical method validation (e.g., limit of detection, inter-assay variability).
  • Confirm normalization to creatinine and units used for reporting.
  • Present both absolute and normalized biomarker changes.
  • Provide effect sizes (Cohen’s d) or 95% CI for each main outcome.
  • Address residual normality violations and possibly re-run analyses with non-parametric methods.
  • Include p-values for within-sex comparisons if available.

The study presents promising pilot data at the interface of nutrigenomics and hepatic metabolism, but substantial methodological clarification and interpretive restraint are required. Addressing the above comments will significantly improve the manuscript’s scientific rigor and credibility.

Author Response

Please find attached- thank you!

Dear Reviewer #2,
Dear Editors,

Thank you for your time, constructive feedback, and the opportunity to revise the manuscript ijms-3934416, entitled “Does Genetic Variation in Detoxification Capacity Influence Hepatic Biomarker Responses to a Liver Support Supplementation Regimen?

In response to your valuable comments, we have provided supporting evidence along with detailed explanations that directly address each point. We hope our responses below, along with the revisions throughout the manuscript, adequately address your concerns. All changes throughout the manuscript have been highlighted in yellow. Additionally, the reference list has been updated based on the revisions, with new references highlighted in yellow.

Kind Regards,
Corresponding authors, on behalf of all authors

Review report 2

The study included only 30 participants divided into two genetic groups (18 LDC and 12 NDC). This sample is underpowered to detect modest genotype × time interactions or sex-specific effects. The authors should perform and report a power analysis or at least acknowledge this limitation more explicitly in the Discussion.

Response: We appreciate the reviewer’s concern regarding statistical power. As a pilot study, our primary objective was to assess feasibility, explore biomarker responsiveness, and identify preliminary trends across genotype groups. The sample size was intentionally limited to allow for controlled implementation and close monitoring of adherence and biological responses. While we acknowledge that this pilot study may be underpowered to detect modest genotype × time or sex-specific effects, our findings offer biologically plausible signals that warrant further investigation in this promising area. We have now provided further explanation of this limitation in the Discussion (lines 391-395), clarifying that these results are intended to inform the design and powering of future, larger-scale studies (including ongoing research currently underway in our lab).

The binary grouping (LDC/NDC) is based on limited polymorphisms (CYP1A1, GSTM1, GSTT1, GSTP1, NQO1). This classification is too simplistic and may not truly reflect “detoxification capacity.” Suggest incorporating or discussing additional relevant genes (e.g., UGT1A, UGT2B, β-glucuronidase activity) and justifying why only these five genes were chosen.

Response: We understand the reviewer’s concern and agree that detoxification capacity is multifactorial and the binary LDC/NDC grouping based on five polymorphisms may provide a limited representation of detoxification capacity. In response, we have expanded the corresponding statement in the Limitations section to clarify the rationale for gene selection and to acknowledge the omission of additional relevant enzymes, including UGT1A/UGT2B isoforms and β-glucuronidase activity (lines 400-404).

The sex difference (higher in females) is interesting but insufficiently explored. The authors could expand the discussion on hormonal regulation of UGT enzymes, microbiome β-glucuronidase, or sex hormone–dependent metabolism to give biological depth.

Response: We thank the reviewer for this insightful comment. We have expanded the discussion of the observed sex difference in D-glucaric acid to include further consideration of hormonal regulation of UGT enzymes, sex hormone–dependent metabolism, microbiome β-glucuronidase activity, and their interactions, providing additional biological depth and context for the observed sex-specific patterns (lines 310-311, 314-317, and 322-323).

The mixture of supplements (glutathione, NAC, α-lipoic acid, DIM, zeolite) is complex. The manuscript lacks justification for their combination, individual dosages, or potential interactions. Include a table detailing doses, proposed mechanisms, and references supporting synergy.

Response: Thank you for this constructive comment. In response, we have designed and added a table summarizing the supplement composition, individual dosages, biological mechanisms, and supporting references, along with providing a justification for the combination used in our study (lines 461-463 and 467-470).

The total sample (n = 30; LDC = 18, NDC = 12) is too small to detect genotype × time interactions with adequate power. Please include a power analysis or explicitly acknowledge this limitation and its impact on negative findings.

Response: Thank you. This comment has already been responded to and addressed in the revised version of manuscript (lines 391-395); please refer to our response to the similar comment above.

The binary classification into “Limited Detox Capacity” and “Normal Detox Capacity” based on only five genes (CYP1A1, GSTM1, GSTT1, GSTP1, NQO1) oversimplifies the complex detoxification network. Provide scientific justification for these markers, and discuss the omission of other critical pathways (UGTs, β-glucuronidase, etc.).

Response: Thank you. This comment has already been responded to and addressed in the revised version of manuscript (lines 400-404); please refer to our response to the similar comment above.

Without a non-supplemented control, biomarker changes cannot be confidently attributed to the intervention. The authors must clearly discuss this as a major limitation and avoid causal wording in the Abstract and Discussion.

Response: Thank you for this valuable comment. This limitation has been explicitly acknowledged in the revised manuscript. We have also carefully revised the Abstract, Discussion, and Conclusion sections to avoid causal wording (lines 33-36, 240-244, and 554-559).

It is unclear whether D-glucaric acid and mercapturic acid values were normalized to urinary creatinine. This step is essential to control for hydration and renal output variability. Please confirm and describe the normalization procedure in the Methods and figure legends.

Response: Thank you very much for this comment. D-glucaric acid and mercapturic acid values have already been normalized to urinary creatinine (nmol/mg creatinine for D-glucaric acid; μmol/mmol creatinine for mercapturic acids). The normalization procedure has now been described in the Methods section and added to the figure legend (lines 488-490 and 136-138).

Several mixed-effects models show non-normal residuals (Shapiro-Wilk p < 0.05). Authors should verify results using non-parametric analyses or data transformation to ensure robustness.

Response: Thank you for this important comment. We agree that residuals for D-glucaric acid, mercapturic acids, and creatinine deviated from normality, and for transparency, these deviations have been explicitly noted in the Results and table footnotes. As you know, linear mixed-effects models are generally robust to moderate deviations from normality, particularly when random effects are included; they also provide unbiased estimates of fixed effects even under mild non-normality. To address this, we have also added a statement in the Discussion highlighting this limitation and advising cautious interpretation of the findings (lines 404-406).

The discussion equates increased urinary levels with improved detoxification capacity. This may instead represent elevated excretion or turnover of conjugates. Revise interpretation and add references differentiating up-regulation vs. clearance phenomena.

Response: Thank you for this insightful comment. We agree with your observation and have revised the interpretation to clarify that increased urinary levels may reflect elevated excretion or turnover of conjugates rather than improved detoxification capacity. We also added supporting references to strengthen this fact (lines 244-247).

The finding that females exhibit higher D-glucaric acid is intriguing but underexplored. Expand on possible hormonal or microbiome-related mechanisms (e.g., UGT2B17 expression, β-glucuronidase activity).

Response: Thank you. This comment has already been responded to and addressed in the revised version of manuscript (lines 310-311, 314-317, and 322-323); please refer to our response to the similar comment above.

Weight and BMI decreased significantly; weight loss can independently affect hepatic enzyme activity. Consider performing an adjusted model or at least discuss this confounder.

Response: Thank you for the thoughtful comment. We’ve addressed this potential confounder by noting that the magnitude of weight and BMI changes was modest and unlikely to fully explain the hepatic enzyme findings (lines 346-349).

The supplement mixture (glutathione, NAC, α-lipoic acid, DIM, zeolite) requires clearer rationale. Include a table summarizing ingredients, doses, proposed mechanisms, and supporting references.

Response: Thank you. This comment has already been responded to and addressed in the revised version of manuscript (lines 461-463 and 467-470); please refer to our response to the similar comment above.

The statement that supplementation “beneficially modulates detoxification regardless of genotype” overreaches the evidence. Rephrase to reflect that effects are potential and preliminary pending confirmation in randomized controlled trials.

Response: Thank you for pointing this out. We have revised the statement to reflect that the observed effects are potential and preliminary, and clarified that confirmation in RCTs is needed before drawing genotype-independent conclusions (lines 384-387).

Figures – Raincloud plots are elegant but busy; consider adding summary bar graphs (mean ± SEM).

Response: Thank you for this thoughtful feedback. While raincloud plots can be information-dense, we believe adding summary bar graphs (mean ± SEM) to the same figure would significantly increase the visual complexity and potentially lead to a redundant representation of the data. Instead of adding a separate graph, we have opted to enhance the clarity and interpretability of the existing plots by extending the figure legend to provide a more detailed description of the figure (lines 134-141).

Terminology – Ensure consistent italicization of gene names (CYP1A1, GSTM1, etc.).

Response: Thank you for this reminder. We have carefully reviewed the manuscript and ensured that all gene names are consistently italicized throughout.

Ethical Statement – Already adequate but add IRB approval date explicitly if available.

Response: Thank you. We have added the date of ethical approval, accordingly (lines 428-429).

Clarity of Hypothesis – Restate hypothesis explicitly at the end of the Introduction, indicating expected biomarker changes.

Response: Thank you for your feedback. We have revised the hypothesis at the end of the Introduction to explicitly state the expected increase in biomarkers (lines 98-101).

  • Add “pilot open-label study” to clarify scope.
  • Specify normalization approach (to creatinine or not).
  • Soften conclusions (replace “beneficially modulates” with “was associated with changes in”).
  • Excellent ethical and biospecimen handling details.
  • Add exact supplement dosages and average adherence rate.
  • Clarify analytical method validation (e.g., limit of detection, inter-assay variability).
  • Confirm normalization to creatinine and units used for reporting.
  • Present both absolute and normalized biomarker changes.
  • Provide effect sizes (Cohen’s d) or 95% CI for each main outcome.
  • Address residual normality violations and possibly re-run analyses with non-parametric methods.
  • Include p-values for within-sex comparisons if available.

Response: We are grateful to the Reviewer for the valuable and constructive feedback. We have carefully revised the manuscript to address the points raised. The majority of the suggestions have been incorporated into the revised version, while we also wish to note that some of the requested details were already available in the original submission and have been further clarified for emphasis.

The study presents promising pilot data at the interface of nutrigenomics and hepatic metabolism, but substantial methodological clarification and interpretive restraint are required. Addressing the above comments will significantly improve the manuscript’s scientific rigor and credibility.

Response: We sincerely thank the reviewer for the positive assessment of the breadth and value of our work. We hope that our revisions and explanations adequately address your concerns.

Thank you once more!